# Characterizing Stream Condition with Benthic Macroinvertebrates in Southeastern Minnesota, USA: Agriculture, Channelization, and Karst Geology Impact Lotic Habitats and Communities

**DOI:** 10.3390/insects16010059

**Published:** 2025-01-10

**Authors:** Neal D. Mundahl

**Affiliations:** Program in Ecology and Environmental Science and Large River Studies Center, Department of Biology, Winona State University, Winona, MN 55987, USA; nmundahl@winona.edu

**Keywords:** biomonitoring, biotic integrity, aquatic insects, stream assessments, stream habitat

## Abstract

Baseline data on stream habitats and aquatic invertebrate communities were needed prior to beginning corrective measures to restore a stream and river system in southeastern Minnesota, USA, impacted by channelization and agricultural runoff. Although >85% of habitat assessments rated stream sites as fair or good, >50% of invertebrate samples exhibited poor or very poor communities. Too many fine stream-bottom sediments (e.g., silt, sand), a lack of habitat diversity, and shallow water resulted in low invertebrate diversity and low numbers of pollution- or disturbance-sensitive species. Communities improved from upstream sites to downstream sites as the river landscape transitioned from a dominance of row-crop agriculture to a more forested drainage with increased coldwater inputs from groundwater springs. Action is needed to reduce channelization disturbances and soil erosion from agricultural fields, especially in the headwaters, before aquatic invertebrate communities can recover.

## 1. Introduction

Biomonitoring of streams and rivers has a long history, dating back at least to the 1850s [1,2,3,4]. Most major biotic groups found in these systems have been used to assess the impacts of human activities on the natural functioning of lotic systems [5,6], from diatoms and other algae [1,7] to aquatic macrophytes [8], benthic insects [9,10,11,12,13], mussels [14], and fish [15,16]. Two common denominators for using these organisms are (1) their general abundance and diversity in natural rivers and streams, and (2) the wide range in tolerance levels among different taxa to various types of disturbance or stress [6]. As streams and rivers become more severely impacted, sensitive or intolerant species predictably decline or disappear from systems, usually expressed as reduced community diversity and/or dominance by the remaining more tolerant species [6].

Although many lotic organisms can be used for biomonitoring purposes, they are not all broadly used for a variety of reasons [6]. For example, some communities may be difficult to sample without using specialized techniques (snorkeling or SCUBA for mussels) or equipment (electrofishing for fish), whereas others may be difficult for non-experts to identify (algae and aquatic macrophytes). Most stream and river biomonitoring efforts have centered around benthic macroinvertebrates and fish due to their ease of collection and their general familiarity to the public [6,16].

Benthic macroinvertebrates have long been the community of choice when monitoring streams and rivers due to several important characteristics [10]. Advantages of using benthic macroinvertebrates include their short life cycles and low mobility that can quickly integrate and express localized environmental conditions, ease of sampling them with limited personnel and equipment, their relatively easy identification, and their high abundance in most systems [10,11]. Because non-scientists can be successfully trained in less than one day to collect and identify common types of stream-dwelling macroinvertebrates, large-scale citizen-based stream biomonitoring efforts make use of macroinvertebrates to assess stream conditions [17].

The objective of this study was to use benthic macroinvertebrates and simple stream habitat measures to assess the stream conditions present at multiple sites in a river system flowing through an agricultural landscape in southeastern Minnesota, USA. This river system contains a mix of warm- and coldwater habitats, channelized and natural channels, and karst geology that facilitates numerous, two-way conduits between surface and groundwaters [18]. The goal was to examine background conditions at each site multiple times over a decade to develop a true picture of the condition of the river system after more than a century of collective human impacts, prior to implementing conservation practices (e.g., reduced tillage, feedlot runoff controls, manure storage, grassed waterways) aimed at alleviating those impacts.

## 2. Study Area

The Root River watershed encompasses an area of 4310 km^2^ in southeastern Minnesota, USA, draining portions of five counties into the Upper Mississippi River [19]. This study was conducted within the upper South Branch Root River drainage basin in eastern Mower and western Fillmore counties, comprising the southwestern portion of the watershed. Only the portion of the South Branch drainage basin (29,919 hectares) spanning from its headwaters downstream through Forestville State Park was examined, representing approximately 7% of the total watershed area of the Root River [19]. In total, 141.1 km of stream were included in the portion of the South Branch drainage examined (Table 1, Figure 1), including both warmwater (maximum summer water temperatures > 24 °C) and coldwater (mean summer water temperatures < 18 °C; able to support sustainable populations of trout) reaches. This sub-drainage was targeted as a priority area for study by the U.S. Natural Resources Conservation Service [20] due to its historically poor water quality (e.g., high nitrates, total phosphorus, total suspended solids, and fecal coliform levels) [18,19] and the strong support of local citizens for improving stream water quality [18].

Land use in the upper South Branch Root River watershed is 90% agriculture, 9% woodland, and 1% other uses (mostly urban) [18]. Streams and rivers within this drainage have been impacted by a combination of channelization and ditching, drainage tiling, soil erosion, agricultural fertilizer and chemical runoff, runoff from livestock feedlots and field-applied manure, and urban runoff [18,21,22]. The eastern portion of the study area is underlain with porous limestone (karst) geology, with numerous sinkholes, springs, caves, and disappearing streams creating many interconnections between surface and groundwaters [18,23]. The South Branch mainstem itself goes underground in the vicinity of Mystery Cave before re-emerging again as a series of springs 2.4 km further downstream [19].

The watershed of the upper South Branch Root River and its tributaries has been impacted by several modifications of the landscape to facilitate agriculture. Seventy-five percent of agriculture lands within the watershed have been drained or tiled, virtually eliminating wetlands from the landscape (Table 1). In addition, beginning in the 1920s [21], 31.1 km of stream (22% of the study area total) has been straightened and channelized to accelerate drainage, including 100% of the headwater tributary now referred to as Judicial Ditch #1 (JD #1) and all (2.1 km) of the upper South Branch upstream from the confluence with JD #1 and 2.5 km below the confluence (Table 1).

## 3. Methods and Materials

### 3.1. Fieldwork

During the summers (June through August) of three time periods (1998, 1999, and 2006/2008), 15 stream sites were surveyed for benthic macroinvertebrates and stream habitat (Figure 1, Table 2). Nine of these sites were located on the South Branch Root River mainstem and six were on streams tributary to the South Branch. Seven of the mainstem sites and two tributary sites were classified as warmwater, whereas two mainstem and the remaining four tributary sites were coldwater (Figure 1). The original survey plan called for only two years of sampling (1998 and 1999), but after major maintenance activities occurred in channelized reaches during 2005–2006, a third survey was conducted to incorporate possible changes related to the channel operations. Only a portion of the sites was able to be sampled during 2006, so the remainder were sampled in 2008.

Benthic macroinvertebrates were collected in triplicate kick-net (D-frame net, 500 µm mesh) samples at each site during each of the three collecting periods. The goal was not to assess whether macroinvertebrate communities changed among time periods, but rather to capture the range of variability that might occur naturally at the sites over an 8-to-10-year time period. Each kick-net sample consisted of invertebrates collected during two, 30 sec kicks from the bottom substrate, one from a fast riffle section and one from a slow riffle section, encompassing a combined area of approximately 0.2 m^2^. Wherever possible, kick samples were collected from three different riffles within the stream site. Kick samples were preserved in 70% ethanol until they could be processed in the laboratory.

### 3.2. Laboratory Work

In the laboratory, all invertebrates in each sample were hand-picked from sample debris with the aid of an illuminated magnifier (10× magnification; Taber Luxo-Lighting, Strathmore, CA, USA). Invertebrates were counted and identified to the lowest practical taxonomic level (usually genus) [24,25,26,27] with the aid of a dissecting microscope (8 to 50× magnification; Nikon SMZ645, Nikon Instruments Inc., Melville, NY, USA). Midge larvae and pupae (Chironomidae) were identified only at the family level.

Habitat at each stream site was examined with two approaches. First, a modification of a simple fish habitat rating index developed for assessing streams in Wisconsin, USA [28] was used during each of the three survey periods. Seven instream and riparian features (bank use, bank stability, pool area, width-to-depth ratio, riffle-to-riffle distance/stream width, silt-free substrate, fish cover) were examined, scored, and used to produce a fish habitat score (range from 0 to 100) and rating (poor, fair, good, excellent) for each site each time period.

The second approach for assessing stream habitat was used only during the 2006/2008 surveys, and involved measuring 18 different habitat variables at each of 12 transects spaced 10 m apart at each site. Variables measured included stream width, water depth, current velocity (at 0.6 depth; Marsh-McBirney Flo-Mate 2000 portable velocity meter) substrate type (boulder, rubble, gravel, sand, silt, clay, detritus), substrate embeddedness with fine sediments (a five-category scale: 1 = <5%, 2 = 5–25%, 3 = 26–50%, 4 = 51–75%, and 5 = >75%) [29], % of stream shaded at noon, % riffle, % run, % pool, % fish cover, riparian buffer width, length of overhanging vegetation, % bare bank, % grassy bank, % forb-covered bank, % shrub-covered bank, % tree-covered bank, and % rock-covered bank. Depth, velocity, substrate type, and embeddedness were assessed at four points on each transect, whereas the remaining variables were measured once on each transect. Substrate categories of sand, silt, and clay were combined and compared to the total substrates assessed to measure % fines present at each site. Water temperature also was measured at each site with a simple bulb thermometer.

### 3.3. Data Analysis

Invertebrate data collected during each survey time period were assessed with a Benthic Index of Biotic Integrity (BIBI) developed for use in regional warmwater and coldwater streams [30,31]. Ten characteristics (metrics) of the invertebrate assemblage collected in each individual sample were calculated and scored to produce a single overall BIBI score and rating for that sample (Table 3). Consequently, three scores were generated for each site for each time period, one for each of the three samples collected during each survey. Separate single-factor analysis of variance (ANOVA; Microsoft Excel) and Tukey’s honest significant difference tests were used to examine possible differences in both sample BIBI scores and sample taxa richness among the 15 stream sites using data from all three survey periods combined. BIBI and taxa richness data were combined across survey periods to increase sample sizes and incorporate the full year-to-year variability in macroinvertebrate communities within the simple comparisons. Simple linear regression (VassarStats: Website for Statistical Computation; vassarstats.net; accessed 3 December 2024) was used to assess the relationship between sample taxa richness and sample BIBI score.

To assess possible relationships between stream physical habitats and benthic macroinvertebrate communities, physical stream habitat scores were compared statistically to total BIBI scores and individual BIBI metric values with simple multiple regression models (JMP Pro 18.0.1; JMP Statistical Discovery LLC, Cary, NC, USA). All correlation coefficients > 0.40 or <−0.40 were judged as indicating strong relationships.

## 4. Results

### 4.1. Habitat Assessment

Habitat rating metrics for the 15 stream sites assessed in the upper South Branch Root River system exhibited considerable variability, but mean values for all seven metrics comprised 50 to 60% of maximum possible values (maximum of 10 for four variables, 20 for three variables; Figure 2). When combined to produce total habitat index scores, a majority (60%) of the stream site habitat index values produced fair ratings (Figure 3). Over 30% of stream site assessments rated the habitat as good or excellent, with <10% rated as poor.

When more systematic habitat assessments were conducted at the stream sites during 2006/2008, additional habitat details became evident (Table 4). Stream sites had highly variable buffer widths (ranging from narrow [<5 m] grassy banks surrounded by row crops to livestock pastures to state park forest lands) and were relatively wide and shallow (width-to-depth ratios around 24:1), with slow current velocities, a dominance of run habitat, and little fish cover. Fine sediments were common and the embeddedness of coarse substrates by fines was high (>50%; Table 4). Although stream banks typically were well vegetated with grass, forbs, shrubs, and trees, bare soil or rocks comprised >20% of total bank observations.

### 4.2. Macroinvertebrate Fauna

Across the three survey periods, 26,760 benthic macroinvertebrates representing 84 taxa were collected and identified from the 15 upper South Branch Root River system stream sites (Table 5). Aquatic insects dominated collections with 70 total taxa, including eight genera of stoneflies, 11 mayflies, 16 caddisflies, 11 beetles, and 14 true flies. Ten insect taxa (including four mayflies, three caddisflies, one beetle, chironomid midges, and a cranefly) and one non-insect taxon (the amphipod *Gammarus*) together comprised >85% of the total individuals collected (Table 5).

Individual benthic macroinvertebrate samples contained as many as 22 different taxa, with the majority (61%) of samples containing between nine and 16 taxa (Figure 4A). When compared among sites, the mean number of taxa/sample generally increased from upstream to downstream (Figure 4B), with some downstream sites (especially those coldwater sites near springs) averaging more than twice as many taxa as other downstream sites. Channelized sites (1, 2A, 2) and those immediately downstream (3AA, 3A) demonstrated the lowest levels of sample taxa richness (Figure 4B). There were significant differences (ANOVA *F*_14,108_ = 4.98, *p* < 0.001) in taxa richness among the sites examined.

### 4.3. BIBI Scores

Ratings of the sample BIBI scores collected from the 15 stream sites indicated that >50% of the macroinvertebrate samples exhibited poor or very poor biotic integrity (Figure 5A). Another 28% of samples were rated as fair, 15% as good, and <6% as excellent. There was an obvious trend of increasing BIBI scores between upstream and downstream sites (Figure 5B), with all upstream sites averaging poor ratings (BIBI score < 30) and downstream sites averaging fair (BIBI score > 30) or good (score > 50) ratings. Channelized sites (1, 2A, 2) and those immediately downstream (3AA, 3A) demonstrated the lowest sample BIBI scores (Figure 4B). Site 12 (Forestville Creek) had taxa richness and BIBI scores more similar to upstream channelized sites than other nearby downstream sites (Figure 4 and Figure 5). There were significant differences (ANOVA *F*_14,108_ = 7.76, *p* < 0.001) in BIBI scores among the sites examined.

By examining sample BIBI metric values (Table 6) in comparison to how those values influenced BIBI scoring (Table 3), it is clear why upper South Branch Root River watershed aquatic invertebrate communities did not rate very well. Mean values for all metrics generally were medium to low on the rating scales. In particular, metrics for the number of Plecoptera taxa, percentage of Plecoptera individuals, percentage of long-lived individuals, number of intolerant taxa, and the number of filterer taxa all were consistently low across the study period (Table 6).

When sample taxa richness values were compared to their specific BIBI scores, a strong positive significant relationship was observed (Figure 6). Although there was pronounced variation in BIBI scores for a given sample taxa richness value (e.g., BIBI score ranging from 10 [poor rating] to 70 [excellent rating] for a taxa richness of 12), overall changes in taxa richness accounted for 59% of the variability in BIBI scores.

In total, 23 habitat variable–BIBI metric pairings (out of 88 possible) produced correlation coefficients > 0.40 or <−0.40 (Table 7). Pool area, width-to-depth ratio, and silt-free substrate demonstrated the most relationships. The number of Trichoptera taxa metric was correlated strongly with four habitat variables plus the overall habitat rating score, whereas the number of intolerant taxa metric was correlated strongly to the habitat rating score and three habitat variables (Table 7). Based on these relationships, benthic invertebrate communities in the upper South Branch watershed exhibited the greatest biotic integrity at stream sites with silt-free substrate, narrow and deep channels, and pool habitats that were neither lacking nor over-dominant.

## 5. Discussion

This multi-year survey of benthic macroinvertebrates and stream habitats in the upper South Branch Root River and its tributaries revealed four important characteristics of this headwater system. First, macroinvertebrate communities generally exhibited poor integrity, having few intolerant taxa and lacking diversity in several important aquatic insect orders (Plecoptera, Diptera, Trichoptera, Ephemeroptera). Second, sites within channelized headwater sections had the poorest aquatic communities compared to groundwater-connected mainstem and tributary sites further downstream. Third, twice as many stream sites had habitat ratings of fair or poor compared to those with good or excellent ratings, with many of the habitat rating metrics exhibiting strong correlations (both positive and negative) with various BIBI metrics. Finally, stream channelization, agricultural land use, and karst geology all have strong influences on stream habitat and aquatic community integrity in this headwater system.

Benthic macroinvertebrate communities of rivers and streams are very representative of the types and magnitude of stressors to which they are subjected [6,9,10,11,12,13]. With a diversity in life cycles, feeding strategies, habitat preferences, and tolerances to varying disturbances or challenges [9,26,27], these communities serve as excellent bioindicators of stream and river health [6]. Several insect taxa in particular (i.e., orders Ephemeroptera, Plecoptera, and Trichoptera, or EPT taxa) respond quickly and predictably to changes in organic enrichment, inorganic pollutants, and stream sediment loading [9,32,33,34,35] often more dramatically than other insect or non-insect taxa [36], and consequently are usually included in some form in most biomonitoring frameworks. The regional BIBI used in this study was developed in and successfully tested on streams and rivers in southeastern Minnesota [30,31] prior to being used to assess macroinvertebrate communities in the South Branch Root River.

The poor and very poor BIBI ratings of most sites examined in this study indicate the level of human-caused impacts experienced by this stream system. Initial impacts were first reported more than a century ago [21], ultimately leading to the joint effort of citizens and governmental agencies in this watershed to implement corrective land conservation efforts to remediate threats to local streams and rivers [18,20]. Even though the numbers of EPT taxa collected overall within the watershed during the study were good (eight genera of stoneflies, 11 genera of mayflies, 16 genera of caddisflies), EPT taxa often were poorly represented in individual site samples (although EPT taxa comprised 7 of the 11 most abundant organisms collected during this study). Typical samples contained no stoneflies, one or two mayfly taxa, and two or three caddisfly taxa, resulting in low scores for several of the specific BIBI metrics. Reduced EPT taxa richness is typical of stressed systems [12,13], as many taxa possess tolerances, lifestyles, and feeding habits that require clean water free of suspended sediments and hazardous chemicals, coarse substrates not embedded by fine sediments, and abundant algal food resources [9,26,27,32,33,34,35]. Overall, 15 of the 35 EPT taxa collected in this study (three genera of mayflies, seven genera of stoneflies, five genera of caddisflies) are classified as “intolerant” taxa [9,26,27], but only two of those genera (*Ephemerella* mayflies, *Brachycentrus* caddisflies) were abundant in the South Branch system, and these were mostly located at the downstream, coldwater sites that had higher BIBI ratings in the good to excellent range. Only a single additional intolerant insect genus (*Nigronia*) and no intolerant non-insect taxa were collected from the South Branch system.

There was a strong dichotomy in macroinvertebrate communities between the channelized headwater sites and the downstream coldwater sites in the South Branch Root River. The significant differences observed among sites in taxa richness and BIBI scores were likely influenced by the differences between these two site groupings. Channelized stream sites often have poor macroinvertebrate assemblages compared to nearby natural channels, due to factors such as habitat homogeneity, silt and sand substrates, and a lack of protective refugia during high flows [37,38,39]. The headwater sites on the South Branch Root River and JD #1 generally had the lowest taxa richness and BIBI scores of all sites surveyed each year, especially after the 2005–2006 channel maintenance work that cleaned out all channelized stream sections in Mower County and removed all trees, shrubs, and other vegetation from the banks. These cleaning operations were permitted by state regulatory agencies, but because the cleaning operations failed to protect bare-soil slopes along the channels and allowed soil to erode into the stream during rainfall, the county and its contractor were fined by the Minnesota Pollution Control Agency [40]. Macroinvertebrate samples collected from channelized sites in 2008 (two years after channel maintenance) averaged only 22 individual organisms and fewer than four taxa each, with some samples containing only a single organism.

By contrast, samples from downstream coldwater sites in natural channels of the South Branch system usually contained hundreds of organisms/sample and up to 22 taxa/sample. The cold, clear water emanating from groundwater springs at or near these sites likely provided good conditions for aquatic insect survival, growth, and reproduction [41,42,43]. Cool, stable thermal conditions associated with groundwater discharges in southeastern Minnesota allow many benthic macroinvertebrate species to increase in abundance [44], producing macroinvertebrate diversity hotspots [43] by providing spatiotemporal refugia against otherwise fluctuating seasonal temperatures and flow conditions [45,46]. Groundwater discharges may allow benthic macroinvertebrates to grow and reach maturity more quickly [47], producing conditions that can promote multi-voltinism and enhanced production in many aquatic insects [46]. The headwaters of the South Branch Root River historically also were dependent on groundwater spring seepage, but those springs were lost to draining, tiling, and ditching in support of agricultural activities [21]. Today, major springs are more limited within the study watershed, with significant spring discharges supporting flows only in Etna, Canfield, and Forestville creeks and the lower portion of the South Branch study area [23]. This historically altered distribution of groundwater springs is likely a major factor in the present condition of benthic macroinvertebrate communities in the upper South Branch Root River watershed.

Most stream habitats in the upper South Branch Root River were in fair or poor condition. This situation is not uncommon within this region [41,48,49,50], where >150 years of agricultural activities have altered the natural landscape and impacted the streams that flow through it [23,51,52]. Row-crop agriculture and livestock production have been linked to stream habitat deterioration via eroding soils, reduced infiltration, altered hydrology, chemical runoff, overgrazing, and livestock manure, resulting in damaged stream riparian areas, excessive channel scouring and/or sediment deposition, channel destabilization, and reduced instream habitat diversity [23,52,53,54,55,56,57,58,59,60,61,62]. Benthic macroinvertebrates are capable of recovering quickly from even catastrophic disturbances [63], but long-term and chronic stressors often associated with agricultural activities within the watershed can eliminate many aquatic taxa and prevent community recovery due to continuing degraded habitats [64]. Many of the metrics used in the macroinvertebrate BIBI in this study were strongly correlated, both positively and negatively, with various stream habitat variables in predictable ways. For example, habitat heterogeneity (i.e., pools not lacking or overly dominant) and silt-free substrate were positively correlated to increased numbers of intolerant, filterer, and caddisfly taxa, whereas high width-to-depth ratios (streams wide and shallow) were correlated negatively to those same BIBI metrics. Overall, with many stream habitats impacted by fine sediment deposits that often smothered the coarser bed materials, and with channels dominated by wide and shallow run habitats, benthic invertebrate communities were limited in both abundance and diversity to those taxa tolerant of the degraded stream habitats.

It may be difficult to easily separate the effects of agriculture, stream channelization, and karst geology on the stream habitats and aquatic invertebrate communities in the upper South Branch Root River system. All three of these factors likely were acting simultaneously to influence stream communities at the various sites examined in this study. The negative effects of stream channelization and other activities associated directly or indirectly with agriculture within the basin were observed most dramatically in the upper headwater reaches, whereas the positive influence of groundwater discharges associated with karst geology were most obvious in the downstream portions of the study area. Multiple factors normally act in combination to create, control, and modify both stream habitats and their biota, and understanding their roles and interactions is an important focus of ecological research in lotic systems [65]. Rehabilitation of streams and rivers in the South Branch Root River drainage will need to target specific localities where conservation practices can be implemented to provide maximum benefits to receiving waters. Past targeted efforts of this kind in the southeastern Minnesota region have experienced somewhat limited success due to insufficient resources to address most of the problematic conditions within the landscape [48]. New state-wide riparian buffer regulations requiring permanent vegetative cover along streams and rivers [50] should jumpstart improvements within the upper South Branch Root River system.

Through 2022, the upper South Branch watershed has been successful in implementing 48 of the planned 102 best management practices (BMPs) on agricultural lands [66]. These BMPs have included grassed waterways, cover crops, grade stabilization structures, manure and feedlot management projects, field-edge prairie strips of perennial native vegetation to intercept runoff, altered cultivation practices, and other activities and equipment designed to benefit local water quality. Across the entire Root River watershed, 716 completed BMPs are estimated to have reduced soil erosion by 11 million kg/year and prevented >17,000 kg of nitrogen and >6000 kg of phosphorus from entering waterways each year [66]. Unfortunately, no systematic surveys of upper South Branch waterways have been conducted to determine if stream habitats and biotic integrity have responded to these reductions in sediment and nutrient delivery. However, periodic maintenance of the channelized headwater sections of the South Branch system to facilitate drainage of this area for agriculture (permitted under current Minnesota regulations) will provide significant continuing challenges to the recovery of these stream reaches.

## 6. Conclusions

Stream habitats and benthic macroinvertebrate assemblages in the upper South Branch Root River drainage in general were in fair to very poor condition. Stream headwaters originally were impacted by historical channelization undertaken more than 100 years ago to drain the landscape for agriculture, with on-going runoff from row-crop fields and livestock operations also currently contributing to stream channel and biotic impairments. Stream habitats and aquatic biota improved further downstream, as the landscape transitioned to a more forested landscape and groundwater impacts stabilized stream temperatures and flows. Rehabilitation of stream environments in this watershed, especially in the headwaters area, will require landowner participation to install and maintain land conservation practices targeted to reduce soil erosion, prevent feedlot runoff, and prevent agricultural chemicals from impacting ground and surface waters. Mandatory statewide riparian buffer rules now in place should contribute at least partially to recovery of instream habitats and aquatic communities.

## Figures and Tables

**Figure 1 insects-16-00059-f001:**
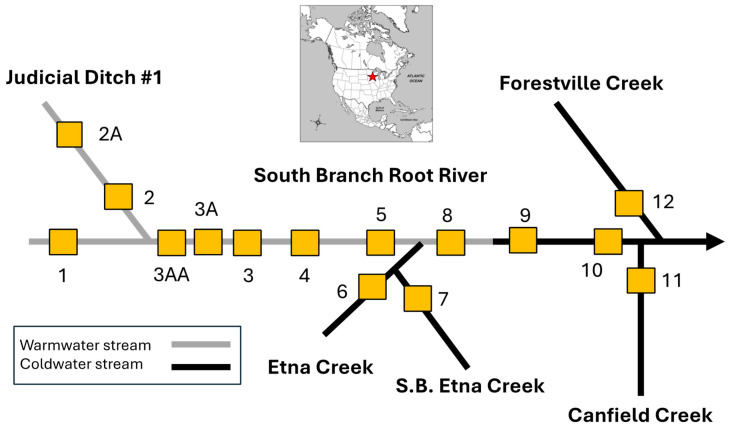
Diagrammatic representation of 15 study site locations on the upper South Branch Root River and its major tributaries. Star on the inset map of North America indicates the location of the study area in southeastern Minnesota, USA.

**Figure 2 insects-16-00059-f002:**
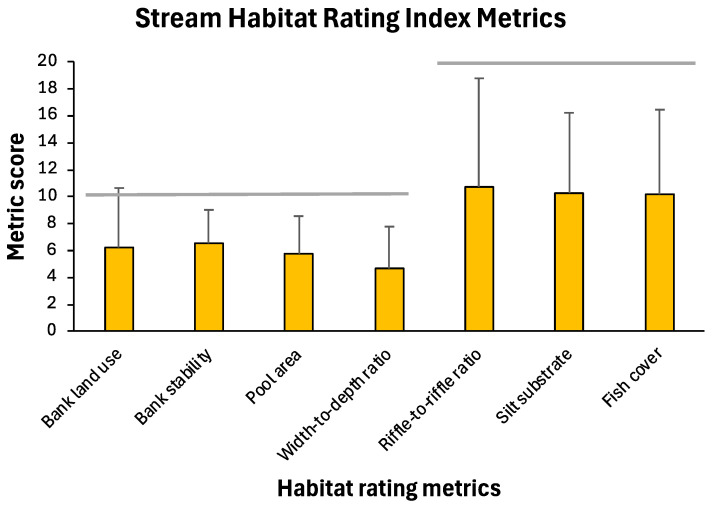
Mean (+one standard deviation) scores for seven stream habitat rating index metrics for 15 sites in the upper South Branch Root River watershed, 1998, 1999, and 2006/2008. Horizontal lines indicate maximum possible scores for the different metrics.

**Figure 3 insects-16-00059-f003:**
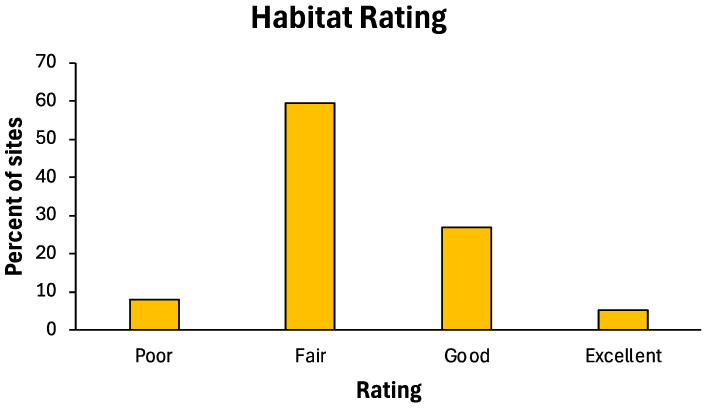
Distribution of stream habitat ratings for 15 sites in the upper South Branch Root River watershed, 1998, 1999, and 2006/2008.

**Figure 4 insects-16-00059-f004:**
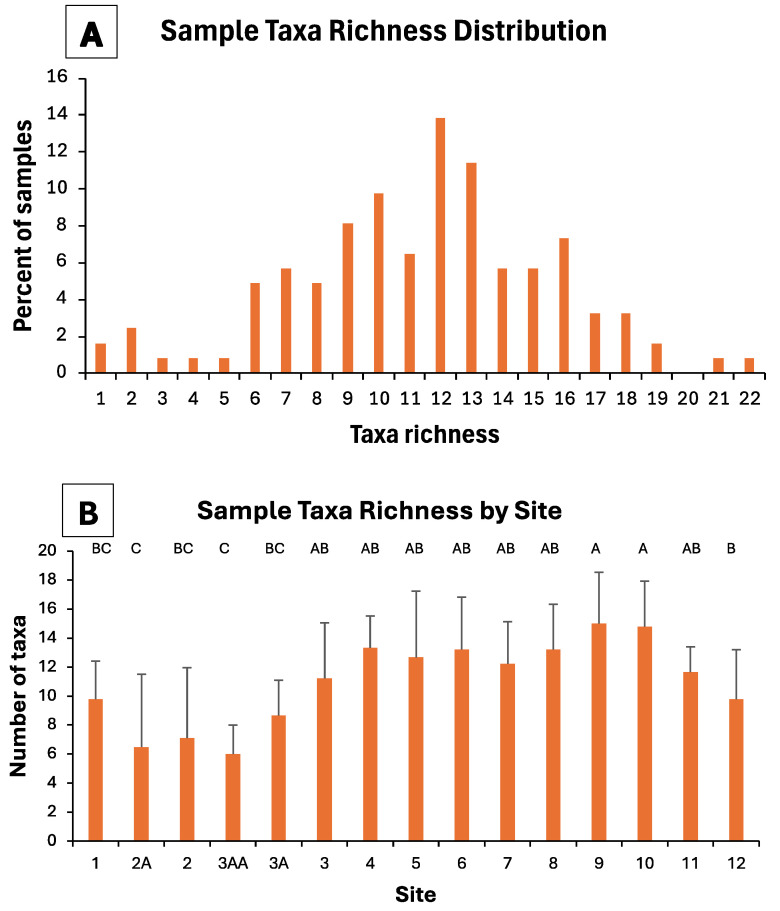
(**A**) Taxa richness distribution of benthic macroinvertebrate samples collected from 15 sites in the upper South Branch Root River watershed, 1998, 1999, and 2006/2008 and (**B**) mean (+one standard deviation) taxa richness of samples from each stream site. In (**B**), bars not having a common letter above them are significantly different from one another (Tukey’s tests).

**Figure 5 insects-16-00059-f005:**
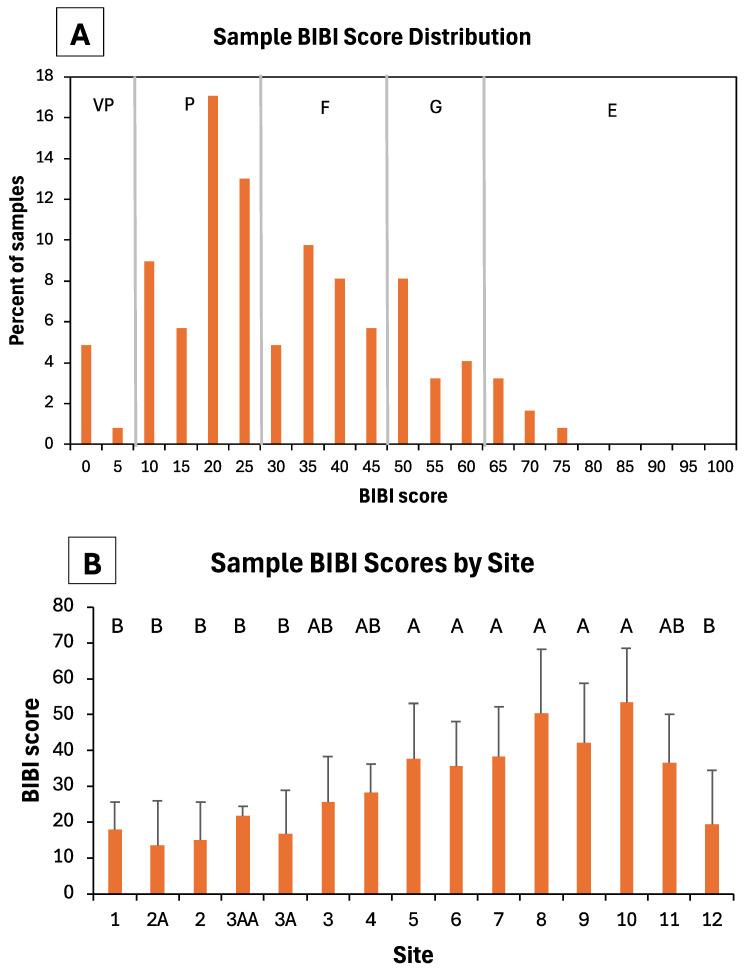
(**A**) BIBI score distribution of benthic macroinvertebrate samples collected from 15 sites in the upper South Branch Root River watershed, 1998, 1999, and 2006/2008. Vertical lines separate scores into five rating classes: VP = very poor, P = poor, F = fair, G = good, and E = excellent. (**B**) Mean (+one standard deviation) BIBI scores of samples from each stream site. In (**B**), bars not having a common letter above them are significantly different from one another (Tukey’s tests).

**Figure 6 insects-16-00059-f006:**
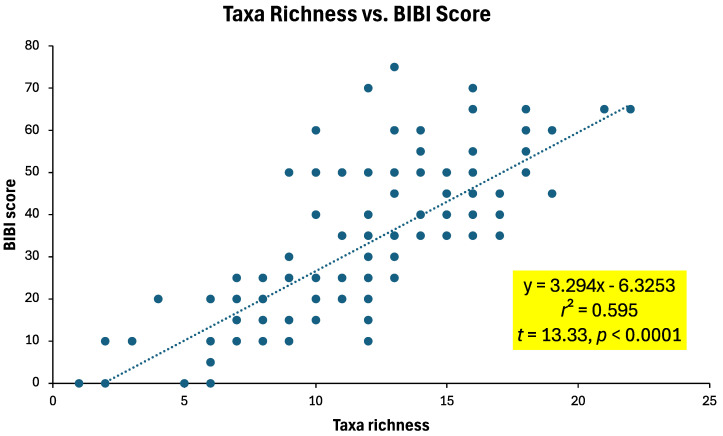
Relationship between taxa richness and BIBI score of benthic macroinvertebrate samples collected from 15 sites in the upper South Branch Root River watershed, 1998, 1999, and 2006/2008. Simple linear regression statistics are included.

**Table 1 insects-16-00059-t001:** Stream statistics for the upper South Branch Root River study area. Data for South Branch Etna Creek are included under Etna Creek.

Variable	S. Br. Root River	Judicial Ditch #1	Etna Creek	Canfield Creek	Forestville Creek
Drainage area (hectares)	13,877	2412	1783	7447	4400
Stream length (km)	77.2	24.1	8.4	27.2	4.2
Percentage channelized	6	100	28	0	0
Average riparian buffer width (m)	30	35	15	20	25
Percentage drained/tiled (agricultural areas)	75	30	40	35	20
Percentage of historical wetlands remaining	<1	<1	<1	<1	10

**Table 2 insects-16-00059-t002:** Benthic macroinvertebrate collection sites on the upper South Branch Root River and its major tributaries in Fillmore and Mower counties, southeastern Minnesota, USA. Collections were made from June to August in 1998, 1999, 2006, and 2008.

Site Number	Stream Name	County	Latitude/Longitude
1	S. Br. Root River	Mower	N 43°38.066′ W 92°31.758′
2A	Judicial Ditch #1	Mower	N 43°39.393′ W 92°32.967′
2	Judicial Ditch #1	Mower	N 43°38.714′ W 92°31.419′
3AA	S. Br. Root River	Mower	N 43°38.776′ W 92°29.215′
3A	S. Br. Root River	Mower	N 43°38.714′ W 92°27.503′
3	S. Br. Root River	Fillmore	N 43°38.160′ W 92°25.762′
4	S. Br. Root River	Fillmore	N 43°37.568′ W 92°24.340
5	S. Br. Root River	Fillmore	N 43°37.601′ W 92°23.426′
6	Etna Creek	Fillmore	N 43°36.060′ W 92°20.706′
7	S. Br. Etna Creek	Fillmore	N 43°36.072′ W 92°20.646′
8	S. Br. Root River	Fillmore	N 43°37.000′ W 92°18.606′
9	S. Br. Root River	Fillmore	N 43°37.555′ W 92°15.985′
10	S. Br. Root River	Fillmore	N 43°37.374′ W 92°13.340′
11	Canfield Creek	Fillmore	N 43°37.320′ W 92°13.468′
12	Forestville Creek	Fillmore	N 43°38.393′ W 92°13.284′

**Table 3 insects-16-00059-t003:** Macroinvertebrate benthic index of biotic integrity (BIBI) metrics and scoring criteria. Life span, tolerance, and feeding metrics based on classification from the scientific literature [24,25,26,27]. Maximum possible score = 100. Total score ratings: excellent = 65–100 points, good = 50–60, fair = 30–45, poor = 10–25, very poor = 0–5.

		Scoring	
Metrics	10 Points	5 Points	0 Points
Number of taxa	>12	7–12	0–6
Number of Plecoptera taxa	>1	1	0
Number of Trichoptera taxa	>3	2–3	0–1
Number of Diptera taxa	>4	3–4	0–2
Number of long-lived taxa	>1	1	0
Number of intolerant taxa	>3	2–3	0–1
Number of filterer taxa	>4	3–4	0–2
Percentage of Plecoptera individuals	>12	6–12	0–5.99
Percentage of predator individuals	>13	6.5–13	0–6.49
Percentage of long-lived individuals	>12	6–12	0–5.99

**Table 4 insects-16-00059-t004:** Summary of stream habitat variables assessed at 15 sites on the South Branch Root River and its major tributaries during 2006/2008. SD = standard deviation, *n* = sample size.

Variable	Mean	SD	*n*
Water temperature (°C)	22	5	15
Stream width (m)	6.17	2.94	180
Water depth (cm)	25	10	720
Current velocity (cm/s)	10	7	720
% Fines	37	34	720
Substrate embeddedness	4.24	0.91	720
% Shaded	29	38	180
% Riffle	19	14	180
% Run	64	19	180
% Pool	17	11	180
% Fish cover	16	32	180
Riparian buffer width (m)	69	82	180
Overhanging vegetation (cm)	30	45	180
% Bare bank soil	16	18	180
% Grassy bank	61	33	180
% Forbs bank	15	14	180
% Shrubs bank	0	0	180
% Trees bank	1	1	180
% Rocky bank	7	14	180

**Table 5 insects-16-00059-t005:** Benthic macroinvertebrate taxa collected from 15 sites on the South Branch Root River and its tributaries, summers 1998, 1999, 2006, and 2008. The 11 taxa indicated with asterisks collectively comprised >85% of total individuals collected.

Scientific Name	Common Name	Scientific Name	Common Name
INSECTA		INSECTA	
**Order Plecoptera**	Stoneflies	**Order Coleoptera**	Beetles
*Acroneuria*	Common stonefly	*Stenelmis* *	Riffle beetle
*Paragnetina*	Common stonefly	*Dubiraphia*	Riffle beetle
*Phasganophora*	Common stonefly	*Optioservus*	Riffle beetle
*Neoperla*	Common stonefly	*Macronychus*	Riffle beetle
*Perlesta*	Common stonefly	*Gyrinus*	Whirligig beetle
*Amphinemura*	Nemourid stonefly	*Agabus*	Predaceous diving beetle
*Isoperla*	Stripetail	*Hygrotus*	Predaceous diving beetle
*Pteronarcys*	Giant stonefly	*Hydrobius*	Water scavenger beetle
**Order Ephemeroptera**	Mayflies	*Sperchopsis*	Water scavenger beetle
*Baetis* *	Small minnow mayfly	*Ectopria*	Water penny
*Pseudocloeon*	Small minnow mayfly	*Psephenus*	Water penny
*Stenonema* *	Flat-headed mayfly	**Order Hemiptera**	True bugs
*Heptagenia*	Flat-headed mayfly	Corixidae	Backswimmers
*Hexagenia*	Burrowing mayfly	**Order Diptera**	True flies
*Ephemera*	Burrowing mayfly	Chironomidae ***	Midges
*Caenis* *	Square-gill mayfly	*Atherix*	Water snipe fly
*Isonychia*	Brush-legged mayfly	*Tipula*	Cranefly
*Ephemerella* *	Spiny crawler	*Dicranota* *	Cranefly
*Tricorythodes*	Little snout crawler	*Hexatoma*	Cranefly
*Paraleptophlebia*	Prong-gill	*Limonia*	Cranefly
**Order Trichoptera**	Caddisflies	*Limnophila*	Cranefly
*Cheumatopsyche* *	Common net-spinner	*Antocha*	Cranefly
*Hydropsyche* *	Common net-spinner	*Pilaria*	Cranefly
*Neureclipsis*	Little red twilight sedge	*Hemerodromia*	Aquatic dance fly
*Polycentropus*	Brown-checkered sedge	*Clinocera*	Aquatic dance fly
*Glossosoma*	Saddle case maker	*Chelifera*	Aquatic dance fly
*Nyctiophylax*	Dinky light summer sedge	*Simulium*	Blackfly
*Brachycentrus* *	Humpless case maker	*Pericoma*	Moth fly
*Micrasema*	Humpless case maker	NON-INSECTS	
*Hydroptila*	Micro-caddisfly	*Physella*	Snail
*Limnephilus*	Northern caddisfly	*Gyraulus*	Snail
*Pseudostenophylax*	Northern case maker	*Helisoma*	Snail
*Heliocopsyche*	Snail case maker	*Ferrissia*	Limpet
*Chimarra*	Fingernet caddisfly	*Sphaerium*	Fingernail clam
*Hesperophylax*	Silver-striped sedge	Hirudinea	Leech
*Psychomyia*	Net tube caddisfly	Aschelminthes	Horsehair worm
*Triaenodes*	Long-horned caddisfly	Oligochaeta	Segmented worm
**Order Megaloptera**	Alderflies, dobsonflies	*Dugesia*	Flatworm
*Sialis*	Alderfly	Acari	Water mite
*Nigronia*	Dark fishfly	*Faxonius*	Crayfish
**Order Odonata**	Dragonflies, damselflies	*Asellus*	Isopod
*Nehalennia*	Sprite	*Gammarus* *	Amphipod
*Ophiogomphus*	Snaketail	*Hyalella*	Amphipod
*Aeshna*	Mosaic darner		
*Calopteryx*	Jewelwing		

**Table 6 insects-16-00059-t006:** Mean (±one standard deviation, SD) BIBI metric values for 15 stream sites in the upper South Branch Root River watershed across all survey periods (1998, 1999, 2006/2008).

	Overall		1998		1999		2006/08	
Metrics	Mean	SD	Mean	SD	Mean	SD	Mean	SD
Number of taxa	11.4	4.1	10.9	2.7	14.1	3.7	9.2	4.2
Number of Plecoptera taxa	0.5	0.6	0.3	0.5	0.8	0.6	0.4	0.7
Number of Trichoptera taxa	2.2	1.4	2.0	1.4	2.7	1.5	2.0	1.4
Number of Diptera taxa	2.8	1.6	2.8	1.0	3.3	1.6	2.4	1.7
Number of long-lived taxa	1.0	1.0	0.8	0.7	1.2	1.3	1.0	1.0
Number of intolerant taxa	1.7	1.6	1.4	1.6	2.0	1.6	1.6	1.6
Number of filterer taxa	2.5	1.4	2.1	0.9	3.1	1.4	2.2	1.4
Percentage of Plecoptera individuals	1.5	3.4	0.8	2.2	1.7	2.1	1.8	4.9
Percentage of predator individuals	7.7	10.5	11.7	10.7	3.5	2.6	8.5	13.2
Percentage of long-lived individuals	3.0	5.2	2.5	3.1	2.3	4.3	4.2	6.9

**Table 7 insects-16-00059-t007:** Relationships of physical habitat variables to benthic macroinvertebrate BIBI scores and metrics at stream sites in the South Branch Root River watershed, based on multiple regression correlation coefficients. Only relationships with coefficients > 0.4 or <−0.4 are shown.

Habitat Variable	BIBI Metric	Coefficient (*r*)
Habitat rating score	Number of Trichoptera taxa	0.540
	Number of intolerant taxa	0.596
	Percentage of long-lived individuals	−0.423
Bank use	Percentage of predator individuals	−0.584
Bank stability	Percentage of long-lived individuals	−0.579
Pool area	BIBI score	0.447
	Number of taxa	0.496
	Number of Trichoptera taxa	0.441
	Number of filterer taxa	0.674
Width-to-depth ratio	BIBI score	−0.437
	Number of Trichoptera taxa	−0.671
	Number of intolerant taxa	−0.583
	Number of filterer taxa	−0.769
	Percentage of predator individuals	0.475
Riffle-to-riffle ratio	Number of Trichoptera taxa	0.530
	Number of intolerant taxa	0.522
	Number of filterer taxa	0.480
Silt-free substrate	Number of Trichoptera taxa	0.459
	Number of intolerant taxa	0.538
	Number of filterer taxa	0.416
	Percentage of long-lived individuals	−0.546
Fish cover	Number of Diptera taxa	0.493
	Percentage of predator individuals	0.556

## Data Availability

Data supporting the reported results are available from the author upon reasonable request.

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
