# Peer review of "Characterizing Stream Condition with Benthic Macroinvertebrates in Southeastern Minnesota, USA: Agriculture, Channelization, and Karst Geology Impact Lotic Habitats and Communities"

_insects, 2025, doi:10.3390/insects16010059_

Round 1

Reviewer 1 Report

Comments and Suggestions for Authors

This is a nice piece of work that I enjoyed reading. The author has investigated the benthic invertebrate communities at 15 sites in a stream system in Minnesota and characterized the stream habitat using two appropriate methods. The invertebrate community data has been used in an index of biotic integrity (BIBI) developed by the author’s research group specifically for Minnesota streams and published in an earlier paper. The BIBI indicated poorer ecological conditions than the physical habitat assessments, the discrepancy being an important part of the discussion. According to the author (Introduction final paragraph) the goal of the study was to develop a “true picture of the condition of the river system...prior to implementing conservation practices aimed at alleviating those impacts”. Whether conservation practices were in fact implemented is not stated but it would be interesting to know given the stream surveys were undertaken about 26, 25 and 16 years ago. If so, have further comparative surveys been undertaken to assess whether conservation practices have had positive effects.  A paragraph dealing with these question s would add to the value of the paper in my opinion.

The paper is particularly well written and the findings are clearly presented in appropriate tables and figures. I have some comments and suggestions, which are presented below. Overall, the Introduction is suitable, the methods employed are appropriate and well described, and the discussion focusses on interesting results obtained in the study.

Specific points

Simple summary. I presume the simple summary is aimed to be read by non-experts in the field. If so, it would be worth making clear what is meant by sensitive species.  Perhaps “pollution” sensitive or something similar could be stated.

Abstract. I suggest deleting “too” from the sentences 5 and 6 lines from the bottom of page 1. Too few sounds like an unnecessary value judgment.

Introduction. The first paragraph is an informative summary of the nature of biomonitoring.

Paragraph 4. What makes the studied river system unique? Is it the combination of factors mentioned in the sentence? It can be argued that stream systems all differ from each other and therefore all are unique!

Last line of Introduction. Have the conservation practices been implemented? See general comments above.

Study area. In addition to giving the buffers trip widths in Table 1, I would like you to expand on their nature and from what the stream is being buffered in the associated text.

Figure 1. Warmwater and coldwater streams are shown in Figure 1. How are these two types of stream defined. I know that the terms are found in the North American stream literature but they are less likely to be familiar to non-American readers/stream ecologists

Methods and Material

The opening sentence of 3.1 Fieldwork is somewhat confusing with respect to 2006-2008 samplings. Can you clarify what was done in these two years (apparently no sampling in 2007) and why both years were sampled?

3.1, three lines from the bottom. Is 0.2 m2 correct? This is only the area of two conventional Surber samplers and seems very small given the amount of sampling effort described.

Page 5, line 3. “...were counted and identified (typically to genus level using references 24-27) with the aid of...”.

Page 5, paragraph 2. Seven instream factors were scored but it is not stated how the scoring was done. Perhaps an instructive table demonstrating the scoring method could be included, perhaps as a Supplementary table.

Paragraph 3. State the type of meter used to measure water velocity.

Paragraph 3, last line. Were these spot temperatures taken at the time of sampling? What was used to measure temperature? State.

3.3 Data analysis

Following statistically significant ANOVAs it would be useful to use Tukey tests (or equivalent) to show where the differences lie, something one wants to know. See specific comments below.

3.3, line 9. It would be helpful to indicate what types of data were combined. Do they include quantitative counts of individuals and presence/absence data of taxa?

Results

I suggest that subheadings be inserted in the Results section to assist the reader. The first of these, something like “Habitat assessment” would be at the beginning of Results. The second would go after Table 4 on page 7 – “Macroinvertebrate fauna”, and the third on page 9 after Figure 4 – “BIBI scores”.

Page 7, line 1 of text. I suggest the first “more” be replaced by “further” as “more” occurs on the following line.

Line 6. After “well vegetated” insert “with grass, forbs, shrubs and trees”.

Page 8. In line 1 I would be inclined to list all 11 taxa that comprised >85% of individuals. Therefore, after “caddisflies, insert “one beetle, chironomid midges and a cranefly” and one non-insect taxon Gammarus”. It is interesting (to me) that oligochaetes are not part of that list given their abundance in many poor-quality streams with fine substrata.

Text below Table 5. I had expected to see some comment about abundance in addition to taxa richness. This is relevant as some of the BIBI variables are based on abundance.

Page 9, line 5. Give Tukey test results here. Figure 4B suggests there maybe two Tukey groups.

Page 10, line 7. As above include Tukey test results.

Page 11, Table 6. The relatively high percentage of predator taxa (ca 10%) except in 1999 is worth a comment.

Page 12, line 1. State the number of sample or add n = to the graph.

Discussion

Page 13, last paragraph. I think you should mention the 2005-6 channel maintenance work in Methods where you explain additional sampling was done. Also make it clear there why sampling was done in 2006 and 2008, not just one of those years.

Page 14. Interesting discussion on this page.

Paragraph 2, line 13. “...were strongly correlated, either positively or negatively, with ...” (In this paragraph and elsewhere it is more correct to say correlated with not correlated to.

Page 15, line 2. Has rehabilitation of streams in the South Branch Root River drainage proceeded? You suggest it should have in the Introduction. I would like to see you explained what action has occurred since the time your surveys were conducted.

Final sentence of discussion. Indeed!

References

Sixty-five references are cited. Although few of them are recent publications, they are all appropriate and do the job satisfactorily.

Reference 2. Book title needs to be in italics.

Reviewer 2 Report

Comments and Suggestions for Authors

Review of: Characterizing Stream Condition with Benthic  Macroinvertebrates in Southeastern Minnesota, USA:  Agriculture, Channelization, and Karst Geology Impact Lotic Habitats and Communities, by Neal D. Mundahl

Overall this is a solid, well-written manuscript.  It is very specific to one region, although the findings and applications likely extend to surrounding areas with similar situations.  In that sense, it serves to bolster what we already know about impacts of anthropogenic disturbance on stream invertebrate communities-in other words, it's not novel in scope but it does add value to the field of study by validating the concepts for this specific region.  I do hope the findings of this paper eventually serve to influence future conservation efforts.  

I do not have any major concerns with the paper, but I offer some comments and suggestions below that I believe will improve it.

1. Simple summary

No comments

2. Abstract

No comments other than you should add the word ‘negatively’ before impact in the last line.

3. Key Words

For key words, don’t repeat words in the title.  Key words are for search engines so there is no need to duplicate.

4. Introduction

Second paragraph, 3rd sentence: strike ‘(invertebrates)’ and ‘(fish)’.  They really are not needed.

I would suggest combining paragraphs 2 and 3 since they address the same subject. And they are somewhat repetitive so maybe reorganize it a bit.

5. Study Area

The inset map in Figure 1 is not particularly helpful other than showing your study sites are in Minnesota.  Perhaps make it larger, or add an actual map of the study site as a separate figure in addition to the diagrammatic representation.

6. Methods and Materials

Table 2.  I suggest adding an additional column to indicate which sites were warmwater or coldwater.

Top of page 5:  I suggest, “Invertebrates were counted and identified to the lowest practical taxonomic level (usually genus)…….” 

Table 3:  For intolerant taxa, how was tolerance determined? Whose tolerance values did you use?

For substrate embeddedness, it would be more meaningful if you reported the mid-points of the value ranges for your analysis.  e.g., 5-25% would become 15%, etc.

7. Results

Figure 2.  If the horizontal line represents the maximum possible score, how can the standard deviation bar for Bank land use exceed that threshold?

Table 4.  For substrate embeddedness, it looks like you used average scale category?  It would be more meaningful perhaps if you used the midpoints of the scale values instead.

Top of Page 9, last sentence:  I suggest something like this—“There were significant differences in taxa richness among the sites examined (ANOVA F14, 108 = 4.98, p<0.001).

Figure 4B.  removed the cross lines in the graph to be consistent with 4A.

Top of Page 10, last sentence: Same comment on structure as indicated above.

Figure 5B.  Eliminate the cross lines.

Page 11, paragraph below Table 6, last sentence, last word: I believe score should be plural as scores.

8. Discussion

Page 13, first full paragraph.  Much of this information is introductory in scope and would be better suited there—except for the last sentence, which does belong in the discussion.  The discussion is really an interpretation of your particular data relative to the published literature.

Also in Page 13, second full paragraph, last sentence just simply say ‘Nigronia’ and strike ‘Order Megaloptera’ since it really is not needed.

9. Conclusions

No comments

10. References

For some references, you provide the issue number (e.g., 6(6)), while others you provide only the volume number.  Please make these consistent—preferably volume number only.
